# Is It Really Home-Based? A Commentary on the Necessity for Accurate Definitions across Exercise and Physical Activity Programmes

**DOI:** 10.3390/ijerph18179244

**Published:** 2021-09-01

**Authors:** Francesca Denton, Sofie Power, Alexander Waddell, Stefan Birkett, Michael Duncan, Amy Harwood, Gordon McGregor, Nikita Rowley, David Broom

**Affiliations:** 1Institute of Health and Wellbeing, Coventry University, Coventry CV1 2DS, UK; dentonf@coventry.ac.uk (F.D.); powers3@coventry.ac.uk (S.P.); waddella2@coventry.ac.uk (A.W.); aa8396@coventry.ac.uk (M.D.); ad5104@coventry.ac.uk (A.H.); ac4378@coventry.ac.uk (G.M.); ac2894@coventry.ac.uk (N.R.); 2School of Sport and Health Sciences, University of Central Lancashire, Preston PR1 2HE, UK; sbirkett4@uclan.ac.uk; 3Department of Cardiopulmonary Rehabilitation, Centre for Exercise & Health, University Hospitals Coventry & Warwickshire NHS Trust, Coventry CV2 2DX, UK; 4Warwick Clinical Trials Unit, Warwick Medical School, University of Warwick, Coventry CV4 7AL, UK

**Keywords:** home-based exercise, definitions, exercise interventions, reporting

## Abstract

*Background*: There is wide discrepancy in how published research defines and reports home-based exercise programmes. Studies consisting of fundamentally different designs have been labelled as home-based, making searching for relevant literature challenging and time consuming. This issue has been further highlighted by an increased demand for these programmes following the COVID-19 pandemic and associated government-imposed lockdowns. *Purpose*: To examine what specifically constitutes home-based exercise by: (1) developing definitions for a range of terms used when reporting exercise and physical activity programmes and (2) providing examples to contextualise these definitions for use when reporting exercise and physical activity programmes. *Methods*: A literature search was undertaken to identify previous attempts to define home-based exercise programmes. A working document, including initial definitions and examples were developed, which were then discussed between six experts for further refinement. *Results*: We generated definitions for universal key terms within three domains (and subdomains) of programme design: location (home-based, community/centre-based, or clinical setting), prescription (structured or unstructured) and delivery (supervised, facilitated, or unsupervised). Examples for possible combinations of design terms were produced. *Conclusions*: Definitions will provide consistency when using reporting tools and the intention is to discuss the issues presented as part of a Delphi study. This is of paramount importance due to the predicted increase in emerging research regarding home-based exercise.

## 1. Introduction

The COVID-19 pandemic has heightened interest in, and identified a need for, home-based exercise research to ascertain its feasibility, efficacy and effectiveness. Globally, most countries have imposed some degree of restriction or lockdown since the emergence of the COVID-19 pandemic. For example, in England the numerous government-imposed lockdowns limited exercise outside of the home to a maximum of once per day for one hour. Therefore, if individuals wanted to engage in more than one bout of exercise per day, at the time of the harshest restrictions within England, home-based would strictly refer to within the home, or the immediate vicinity of, such as the garden and/or driveway. Additionally, in extreme circumstances such as in Italy, individuals were unable to leave their home entirely or were restricted to hotel quarantine. At the time of writing some countries’ lockdowns are ending such as in Australia and the USA.

A recent systematic review concluded that across all populations (apart from those with eating disorders) physical activity behaviour decreased, and sedentary behaviour increased after COVID-19 restrictions were implemented [1]. It is therefore important to accurately label home-based exercise programmes addressing this change in behaviour, in order to efficiently identify suitable interventions.

The reporting of exercise interventions lacks consistency when using the term home-based. This is demonstrated by studies which have fundamentally different designs, frequently being labelled as home-based, which makes searching for relevant literature challenging and time consuming. To highlight one of the many inconsistencies, a range of locations have been used to describe home-based including programmes restricted to within the home, the community, or a combination of both, at the discrepancy of the participants [2]. We contest whether exercise or physical activity programmes should be defined as home-based if the physical activity behaviour is not taking place in the home or the immediate vicinity.

It is important to acknowledge that multiple reporting tools for a variety of exercise programmes have been generated such as the Consensus on Exercise Reporting Template (CERT) [3] and the proposed Physical Activity Scheme (PARS) taxonomy [4]. However, the terms that are utilised within these tools are not consistently defined, creating a reliance on an assumed mutual understanding. This assumption may provide an explanation for the inconsistencies and misuse of the term home-based exercise within the published literature. Looking specifically towards defining and reporting home-based programmes, a preliminary literature search highlighted several reporting systems for interventions and programmes. Lopez et al. [2], suggested five reporting items: (1) location(s) of exercise, (2) supervision, (3) behavioural supports/resources, (4) technology and (5) deviations. Another proposed reporting method uses a simple coding system, for example ‘SGCP30x3′ would refer to a Supervised Group exercise programme in the Community prescribed by a Physiotherapist for 30 min three times a week [5]. Both reporting tools utilise the terms supervised, community and home-based, which are inconsistently defined across the exercise science research and physical activity and health field due to changes over time from influences such as technology. Hence the individual researcher or practitioner’s perceptions may bias the use of these terms.

Attempts have been made to provide clarification in exercise oncology, where the term independent exercise was coined to replace home-based exercise [6]. This may generate confusion as home-based interventions often compare participants to those who are told to go about their normal activity, which could be misconstrued as independent exercise. This further highlights the need for an accurate definition of home-based and consistent reporting of exercise and physical activity programmes, allowing for easier literature searching, protocol comparisons and, replication for future real-world exercise programme development. Therefore, we propose a clearer more coherent definition of home-based. This in turn highlights the need to define other terms to support the existing reporting systems identified, for clearer reporting of exercise programmes across three domains (see Figure 1).

## 2. Domains to Consider

### 2.1. Location

Across the literature, the phrase home-based has been used to describe exercise programmes in a plethora of locations, ranging from exercise undertaken inside the home, outdoor spaces and gardens, to group activities in a community hall [2]. Exercise programmes undertaken outside of a clinical setting, whether that be in the home or community, are often classified as home-based by default. Whilst this may be effective in distinguishing between clinical and non-clinical, the external environment to a clinical setting is widely variable. It provides exposure to factors such as social interaction, the neighbourhood environment and green space, all of which can influence enjoyment, engagement, and adherence to exercise [7,8,9]. Therefore, the term home-based cannot simply be considered a blanket phrase for exercise programmes outside of a clinical setting, its use should be accurate and consistent across all programme descriptions and the location should be specified.

During the harshest COVID-19 restrictions in England, commencing March 2020, if individuals wanted to exercise more than once per day, this had to be within their home, or in the immediate vicinity of, further warranting a more narrowed specification of what home-based exercise is. Hence, the identification of three locations: (1) home-based, (2) community/centre-based and (3) clinical setting (see Table 1) to distinguish between exercise programmes is warranted, to add specificity to descriptions, and to allow for a level of comparison that is not currently available in the literature.

### 2.2. Prescription

Some exercise programmes provide generic physical activity or exercise guidelines to the participants, such as the World Health Organisation’s Guidelines on Physical Activity and Sedentary Behaviour [10]. However, these recommendations are not individualised, and often include signposting to alternative resources or advice that can be accessed within the public domain. However, one size does not fit all, and more personalised and individualised programmes are essential for some populations such as those with diagnosed clinical conditions or older adults. These programmes often involve a predetermined exercise prescription strategy for each participant. To tailor the programme to a person’s health condition or activity goal, an example would be setting individual step goals in a pedometer-based intervention. Therefore, we identified two programme designs: (1) structured and (2) unstructured (see Table 1). This allows for differentiation between individualised programmes and those that solely provide guidelines.

### 2.3. Delivery

It has been previously suggested that hospital exercise programmes are distinguished from home-based alternatives by the degree of supervision, coining all programmes outside a centre as unsupervised [2]. The rising interest in home-based exercise has increased the need to explore ways to maximise adherence—a common pitfall of traditional supervised exercise [9]. Thus, supervision is becoming an important characteristic which may influence adherence to home-based programmes. This tends to rely on printed materials and scheduled check-ins to assess compliance with the programme and address any barriers [2]. Usually, this is with qualified health and/or exercise professionals, akin to the supervised prescription of hospital programmes. Therefore, the location of the programme should not immediately determine whether a programme is supervised or unsupervised.

Across the literature there are myriad descriptions of supervision in home-based interventions. The degree of supervision can range from directly monitoring the exercise in person, to facilitating a programme with scheduled check-ins, to an absence of any supervision. Frequency of in-person check ins can range from weekly to monthly, and more recently, remote check ins via telephone or video calls have been implemented [2,11,12,13,14]. The COVID-19 pandemic has not only highlighted the need for this remote option, but also participant receptivity to such methods by alleviating for example time and travel burdens. The quantity and quality of support provided by supervision will therefore vary and may influence adherence.

We suggest a clear distinction should be made between exercise that is: (1) supervised (2) facilitated or (3) unsupervised (see Table 1), and that the degree of supervision should be clearly stated in the programme description.

## 3. Demonstrating Use of the Definitions

To put these definitions into context, Table 2 provides examples of possible subdomain combinations. In addition to Table 2, we recognise that more complex scenarios require further explanation. For example, a residential care home could be perceived as either a community- or a home-based setting by different stakeholders. When an exercise instructor leads a seated exercise class with residents in a communal space within a care home, this would be classified as a community-based, supervised and structured exercise programme open to the social influence of other group members. However, if an individual in a care home wanted to increase their physical activity and began walking daily in the care home garden, this would be classified as a home-based, unsupervised and unstructured activity.

With the emergence of new technology, it is important to understand how gamified applications would be classified. For example, Pokémon Go™ [15] inadvertently increased physical activity (in some cases) in a communal setting, despite this not being a specific aim of the application. Thus, this would be classified as community-based, unstructured, and unsupervised exercise. Applications that use this technology to specifically improve health with goals/challenges could be seen as more structured and/or facilitated. More established forms of technology such as exercise DVDs or smart home gym equipment such as Peloton™ would be classified as home-based, structured, and unsupervised, as it has a set structure without an exercise professional present. Although instructors may be able to provide motivation during a class, they are often unable to see you through video calling to correct form and technique.

Wearable activity devices should be in most cases considered as a tool to monitor or deliver exercise interventions, as a healthcare professional would have designed the exercise programme. A physical activity monitor may be used to aid a researcher in facilitating an exercise programme rather than the monitor facilitating the programme itself.

In cases where programmes may satisfy multiple definitions, it is important to state each design. For example, if someone attended a gym class three days per week and undertook one Peloton™ bike session per week at home, the main definition would be community-based, supervised and structured, with a supplementary home-based, structured and unsupervised exercise element.

## 4. Conclusions

The term home-based is largely used inaccurately and inconsistently within the published literature. Its current use does not help the science of physical activity, exercise, or behaviour change. It is time for a revised set of definitions which will enable accurate summary of, and thus comparison between, exercise programmes with the accurate use of universal key terms. A wide adoption of these terms will aid the undertaking of reviews across a broad spectrum of population groups. The process will be less time-consuming with more accurate search results when utilising stringent inclusion and exclusion criteria. We recommend that future research should describe their exercise interventions using the definitions stated above, in order to provide clarity and consistency, and to propose a universal definition for home based as follows:

Exercise or physical activity (excluding ADLs) undertaken inside or within the immediate vicinity of the home (including the garden and/or driveway).

Accurate reporting will improve the overall quality of literature regarding exercise and physical activity interventions [2,5]. Following the COVID-19 pandemic, we anticipate further increases in research investigating the effectiveness of home-based exercise programmes, prompting a greater need for specific definitions to ease comparisons across emerging literature within the next few years. We acknowledge that as restrictions ease people may return to community-based exercise facilities. However, we believe there will be an ongoing demand for home-based physical activity and exercise programmes in the future. Reasons for this may include: personal preferences, time constraints, childcare arrangements, cost and for people who are unable to commute to exercise facilities, or living in rural areas where opportunities may be more limited. At the time of writing, some countries are still under harsh restrictions with no guarantee of permanent easing due to potential new strains of COVID-19. There is always the possibility for future restrictions which further emphasises the importance of researching and understanding home-based exercise and appropriately classifying it.

Whilst there is an understanding that these definitions will not be applicable to all exercise programme locations, prescriptions, and delivery methods, it does however, provide a tool applicable to the vast majority. To ensure that this tool remains relevant it will need to be reviewed and revised over time, to allow for full inclusivity of all types of exercise programmes. For example, an increased use of technology to monitor and prescribe exercise [16] may identify the need to generate a completely new subdomain, such as exercise prescription using artificial intelligence. Overall, these definitions will provide consistency when using existing tools to report exercise programmes and the intention is to discuss the issues presented in this commentary as part of a Delphi study.

The authors welcome dialogue with the exercise science community and physical activity and health field and for any comments or suggestions the reader is encouraged to contact the corresponding author.

## Figures and Tables

**Figure 1 ijerph-18-09244-f001:**
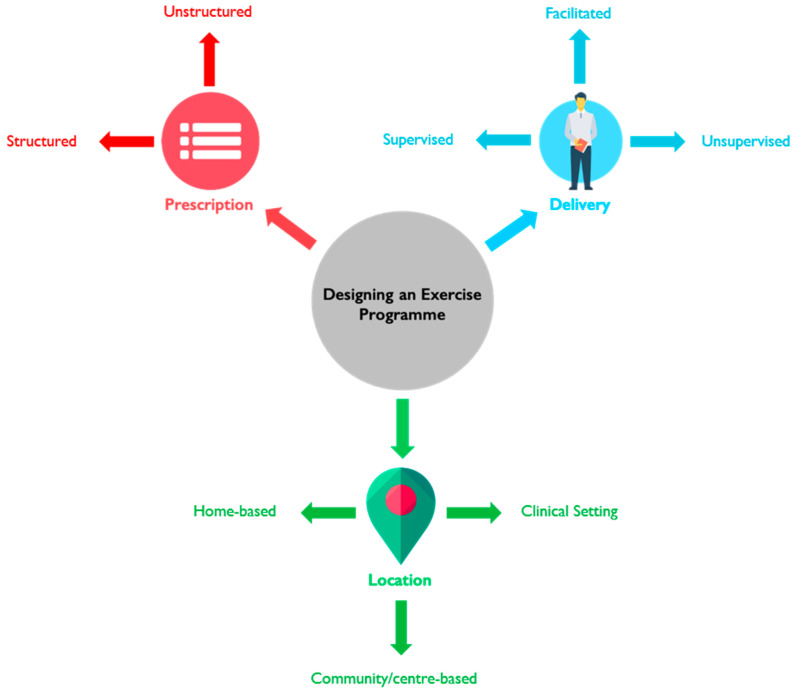
Domains and subdomains that require accurate definitions when reporting an exercise programme that are commonly utilised within existing reporting guidelines [3,4].

**Table 1 ijerph-18-09244-t001:** Definitions for exercise and physical activity programme subdomains.

Programme Domain	Description	Definition
**Location**	Home-based	Exercise or physical activity (excluding ADLs) undertaken inside or within the immediate vicinity of the home (including the garden and/or driveway).
	Community/Centre-based	Exercise or physical activity undertaken in a public open access setting such as a green space, leisure facility, gym, or a community centre.
	Clinical Setting	Exercise or physical activity undertaken in a clinical setting, such as a health care facility.
**Prescription**	Structured	An exercise or physical activity programme based on the FITT principles and tailored to specific health and fitness goals.
	Unstructured	Exercise or physical activity that includes ADLs which have not been specifically prescribed by a healthcare or exercise professional.
**Delivery**	Supervised	Exercise or physical activity undertaken in the presence of a healthcare professional or qualified fitness instructor, either virtually or in person to ensure safety and or correct technique.
	Facilitated	Exercise or physical activity undertaken without the presence of a healthcare professional or qualified fitness instructor but with scheduled meetings or check-ins between sessions to monitor progress and provide support (virtually or in person).
	Unsupervised	Exercise or physical activity undertaken without the presence of a healthcare professional or qualified fitness instructor. No support or progress tracking appointments scheduled (virtually or in person).

Activities of Daily Living (ADL) Frequency, Intensity, Type, Time (FITT).

**Table 2 ijerph-18-09244-t002:** Case study examples.

Subdomain Definition Combinations	Example
**Home-based**	
Home-based, structured, supervised	A circuit programme undertaken at home with an instructor monitoring the exercises via live video call.
Home-based, structured, unsupervised	An individual with sarcopenia undertaking a range of strengthening exercises in their living room alone. The exercises have been prescribed and taught by an instructor before the programme began.
Home-based, structured, facilitated	An individual embarks on a 12-week strength training programme using objects found at home, with weekly phone calls from the instructor to monitor progress and assist with goal setting.
Home-based, unstructured, supervised	An individual living in a care home carrying out walking exercise in the garden supervised by a health care assistant.
Home-based, unstructured, unsupervised	As a result of COVID-19 restrictions, an individual undertakes exercise in their living room without following a specific programme.
Home-based, unstructured, facilitated	Following a health screening, a person is given advice to increase their physical activity at home, which will be followed up on at a future screening appointment.
**Community/Centre-based**	
Community/centre-based, structured, supervised	Group exercise class in a community hall led by an instructor.
Community/centre-based, structured, unsupervised	Walking undertaken outside a place of residence individually or with family/friends at a pre-determined, monitored, intensity.
Community/centre-based, structured, facilitated	Accessing a community gym to individually follow an exercise programme set by a PT outside of the usual scheduled sessions.
Community/centre-based, unstructured, supervised	Children’s soft play activity in a community hall overseen, but not led, by an adult.
Community/centre-based, unstructured, unsupervised	Meeting a friend to go for a bike ride around the neighbourhood.
Community/centre-based, unstructured, facilitated	Following an appointment with a healthcare professional, an individual starts to increase their physical activity behaviour by exploring the public green space outside of their home. They will report this at their next appointment with the HCP.
**Clinical Setting**	
Clinical setting, structured, supervised	A 12-week cardiac rehabilitation programme that takes place in a hospital, is prescribed individually and is supervised by a clinical exercise specialist.
Clinical setting, structured, unsupervised	The authors are not aware of a circumstance where exercise without supervision would be undertaken in this context.
Clinical setting, structured, facilitated	The authors are not aware of a circumstance where exercise without supervision would be undertaken in this context.
Clinical setting, unstructured, supervised	Rehabilitation patients may attend a rehabilitation centre to carry out exercise which they enjoy. During this time there is a HCP available to answer questions and help patients with exercise cues and correction but is not providing structured programming.
Clinical setting, unstructured, unsupervised	The authors are not aware of a circumstance where exercise without supervision would be undertaken in this context.
Clinical setting, unstructured, facilitated	The authors are not aware of a circumstance where exercise without supervision would be undertaken in this context.

Personal Trainer (PT), Health Care Professional (HCP).

## Data Availability

Data sharing not applicable. No new data were created or analysed in this study. Data sharing not applicable to this article.

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
