# Peer review of "Is It Really Home-Based? A Commentary on the Necessity for Accurate Definitions across Exercise and Physical Activity Programmes"

_ijerph, 2021, doi:10.3390/ijerph18179244_

Round 1
Reviewer 1 Report
I really enjoyed reading your manuscript. I strongly believe that a correct classification is needed to cluster the exercise administration correctly. Your paper will guide the research in the field of physical activity.
Just two remarks:
1) Would be interesting to add another figure about Table 1, readers could be more attracted by it (just a suggestion, not mandatory).
2) Remeber to write the final part of the manuscript: Institutional Review Board Statement etc.
Reviewer 2 Report
Thank you for the opportunity to review this commentary paper for your reputable journal. the commentary paper seems to be averagely fine but lacks conciseness and precisions that merit the commentary article. Therefore, the author needs to rework this paper in conjunction with the experts in the field of exercise science.
"See my comments on the article attached"
a.) In line 24: is the purpose of this commentary only for the proposed Delphi study? or for other subsequent research when using the definitions? pls, clarify.
b.) In line 41: with this statement, it seems the author is trying to conduct a home-based exercise intervention study. i.e. how effective is a home-based exercise program?
c.) In line 66: I am not sure about this saying that "some definitions are not scientifically accepted". I do not agree with this statement. although, the definitions might be a better definition at a time than another. This depends on Lopez's assumption.
d.) In line 113: this statement looks like a standalone statement and does not seems to make any differences. It may be deleted or make it a continuing statement along with the statement on activity goals
e.) General comment: In my opinion, the commentary paper needs to be critic and revised. There are also seems to be a number of repetitions and does limit the flows. Organization and rearrangement of the text are also necessary in order to increase easy readability and flow. At some point, the authors were not following their purpose of the study which create instability during the writing, this needs to be avoided. Since the commentary is open to dialogue, the author should avoid using some language that has not yet been scientifically established when citing examples as did in some parts of this commentary paper.

Reviewer 3 Report
I’m very grateful for the opportunity to review this manuscript, and I hope that my comments will help the authors to improve the current state of the document.
The situation caused by COVID-19 has changed the habits and behavior of people when it comes to physical activity and / or physical exercise. Obviously as a consequence of the restrictions (each country has had different restrictions and the authors only speak of England and Italy), many people have carried out physical activity at home, as it is the only possible option in a large number of countries.
However, the end of restrictions and the possibility of people's mobility has largely ended the practice of physical activity at home. In this sense, many people have returned to the practice in specialized centers (fitness centers) and a large population engages in outdoor activity. As a consequence, connections to online activities offered both by specific channels (specialized personal trainers via YouTube, Instagram, etc.) and by specialized centers have dropped to the point that many centers have canceled online activities.
In my opinion, the authors should first carry out a review in the literature of the number of articles published focused on physical activity at home, so that it can be verified that the main problem of the study is really a situation that will have a path in the future.
I consider, from my point of view, that this document does not have the necessary depth for publication in its current state. Talking about terminology requires a thorough review of the original explanatory theories and constructs, but this has not been done.
Finally, I have doubts about the suitability of this manuscript in the International Journal of Environmental Research and Public Health. For this reason, I ask the authors to identify which areas are aligned with this manuscript after reviewing the "Aim & Scope" section, because in my opinion this type of document is more coherent in a journal focused on psychology.
In the sections "Institutional Review Board Statement" (lines 209-216) and "Informed Consent Statement" (lines 217-219), the authors must present one of the options.
I recommend that the authors review the publication rules in relation to the references: the number of the journal is not included, only the volume; doi not included.
Round 2
Reviewer 2 Report
None
Author Response
There were no identifiable comments to respond to